# Effect on catch efficiency and bycatch by introducing an Excluder device in the trawl fishery for lesser sandeel (*Ammodytes marinus*)

**Ole R. Eigaard**[1]☯*, **Claus R. Sparrevohn**[2]☯, **Mathias Søgaard**[1], **Bent Herrmann**[1,3,4]☯

1 National Institute of Aquatic Resources (DTU AQUA), Technical University of Denmark, Kgs. Lyngby, Denmark, 2 Danish Pelagic Producers Organisation, Copenhagen, Denmark, 3 SINTEF Ocean, Trondheim, Norway, 4 UiT the Arctic University of Norway, Tromsø, Norway

☯ These authors contributed equally to this work.
* ore@aqua.dtu.dk

**Data Availability Statement:** All relevant data are within the manuscript and its Supporting Information files.

## Abstract

Sampling of the North Sea trawl fishery for lesser sandeel (*Ammodytes marinus*) showed that 96% of the catch weight consisted of the target species, and experimental sea trials demonstrated that the observed small bycatch percentages of haddock (*Melanogrammus aeglefinus*), mackerel (*Scomber scombrus*) and grey gurnard (*Eutrigla gurnardus*) could be significantly lowered by inserting a netting-based sorting device, an Excluder, in front of the codend. The sandeel fishery is conducted with small meshes in the codend, due to the small size and elongated body shape of this species. It is not mandatory for sandeel trawls to have any other selection devices than the small-meshed codend, and this can potentially cause problems with bycatch of unwanted species, if these are abundant on the fishing grounds. Therefore, we sampled the catch composition in this fishery and further, we compared the capture efficiency and species composition of a standard trawl, and one fitted with an additional sorting device called the Excluder. Overall, results showed small percentages of bycatch in the trips sampled and during the trials, the excluder significantly reduced the bycatches of mackerel, grey gurnard, and haddock above certain sizes. For other bycatch species results were inconclusive due to wide confidence limits affected by low bycatch numbers during the trials. The overall capture efficiency for the target species was not affected by adding the excluder in the trawl except for a significant reduction for a few semi-centimetre groups of the largest sizes of the species. These results highlight the potential of the Excluder as a bycatch reduction tool in the sandeel fishery for situations where bycatch can constitute a problem.

## Introduction

### Stock characteristics and resource exploitation

The lesser sandeel (*A. marinus*) is an important forage fish species in the Northeast Atlantic. It spends most of the year buried in the bottom sediment and resurfaces during spawning

**Funding:** We wish to thank the Ministry of Food, Agriculture, and Fisheries of Denmark (https://fvm.dk/) for funding the work presented here through the GUDP (Green Development and Demonstration Program) in the project 'MiniMakS' (34009-20-1674). The funder did not play any role in the study design, data collection and analysis, decision to publish, or preparation of the manuscript.

**Competing interests:** The authors have declared that no competing interests exist.

around 1$^{st}$ of January and during feeding in spring and early summer [1, 2]. It feeds mainly on copepods, but amphipods, mysiids, and juvenile sandeel can also form part of the diet [3]. When feeding, the sandeel aggregate in large, dense shoals just above the seabed and the shoals exhibit a strong herding reaction to the front part of trawl gears [4]. Consequently, the species can be fished with large, herded-volume (HV) trawl types [5], which enable very high catches per unit of effort (CPUE) and profitable fisheries even if the fish price is generally low. Historically, the North Sea sandeel population has supported a large commercial fishery. The fishery was developed in the 1970s and catches peaked between mid-1980s and mid-1990s with landings of up to 1.2 million tonnes [6]. In the early 2000s the stock biomass declined significantly, possibly related to a regime-shift in the North Sea [7, 8] and since then yearly landings have fluctuated below 500.000 tons [6]. The Danish trawl fishery targeting lesser sandeel (*A.s marinus*) is conducted with a small-mesh codend due to the morphology of this species (small size and elongated body shape). The standard trawl for this fishery does not contain any other selection devices than the small-meshed codend and although the overall fish bycatch percentage is estimated to be only 2% [9], the total volume caught is generally high and this can cause problems with bycatch of unwanted species being simultaneously abundant on the sandeel fishing grounds. However, since 2020 the fishing industry has tested and increasingly implemented a sorting system called the 'Excluder', which has proven efficient in sorting out unwanted fish bycatch in the small-meshed trawl fishery for Norway pout (*Trisopterus esmarkii)* in the North Sea [10].

## Experimental objectives

In this study we examine the potential for the Excluder sorting system to improve the sustainability in the North Sea sandeel fishery. The Excluder represents an industry-driven development of a sorting grid alternative, which has been designed by Greenline Fishing Gear. The Excluder has previously proven successful for bycatch reduction in a similar type of trawl fishery [10] and is essentially a 30-meter lined tube, characterized by being flexible and made only of netting and PVC without any rigid materials. This innovative design is particularly beneficial for net drum users, as it ensures the Excluder can be easily reeled onto net drums, enhancing both operational efficiency and safety. Furthermore, since the Excluder is designed with a considerably larger selection area than that of rigid sorting grids, it is hypothesized that the Excluder will also have a more efficient size-based separation of target and bycatch species compared to a grid sorting system; i.e., enable a more optimal trade-off between maximizing release of unwanted bycatch and minimizing loss of target species. Here we analyse the potential of the Excluder to improve the selectivity and sustainability in the Danish sandeel fishery in an alternate-haul trawl experiment, where the Excluder is tested against a standard trawl as used in the Danish fishery. The experimental fishery was conducted from the commercial 70 m trawler "S205 Ceton" in the North Sea in May 2022, with the overall objective to examine the catch composition in the Danish lesser sandeel fishery and to quantify the effect on capture efficiency and species composition from using the Excluder.

## Materials and methods

### Fishing vessel, fishing grounds, and gear

We conducted the experiment on board the 70 m long pelagic trawler "Ceton S205" from 11 to 24 May 2022, on the lesser sandeel fishing grounds West of Denmark (Fig 1). The trials were conducted as single trawl experiments with the same trawl fished alternately with and without the Excluder mounted. All fish was caught for commercial purposes and no fish was

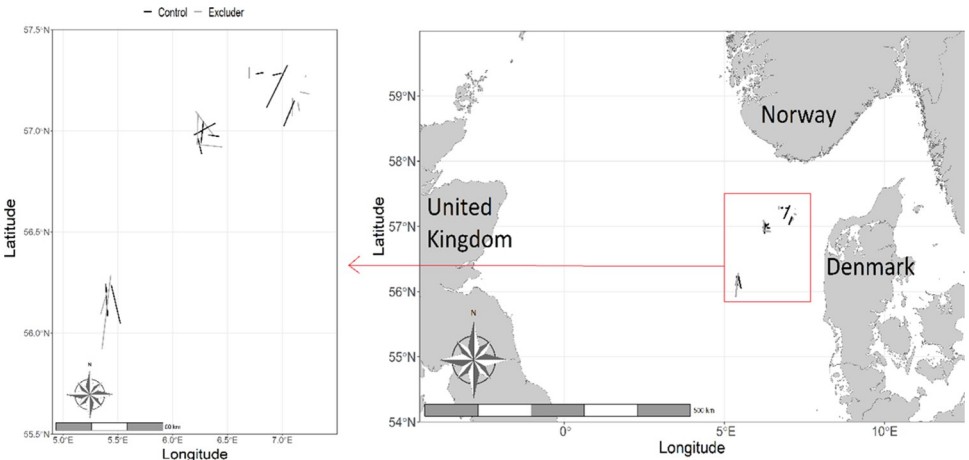

**Fig 1. Spatial distribution of the 21 experimental trawl hauls.** The black lines are hauls with Excluder and the grey lines are hauls without. Note that all hauls are presented as straight lines between the individual start and end positions, even if the actual trawl tracks generally have a much less regular pattern.

anesthetized or sacrificed for the purpose of this study. No formal waiver of ethical approval was required to conduct this study on fish, as all sampling was done on commercial catches of fish.

## The Excluder bycatch reduction device

The Excluder is an all-net section with two PVC kites, inserted as an extension piece of the trawl, which consists of a 30-meter outer-net part and an 11-meter inner selection tube. The inner tube is essentially cone-shaped with an outlet in a bottom panel of the 30 m extension piece, just before the codend. To reach the codend, fish must pass through the meshes of the inner selection tube and continue along the outer panel to the codend (Fig 2). The outer-net of the extension piece was made in diamond meshes with a mean mesh opening of 16 mm. The inner selection tube was made from knotless netting (square meshes) with a mean full mesh opening of 62.8 mm (SD: 1.7 mm). The mean full mesh opening of the codend was 6.35 mm (SD = 0.20). The mesh measurements were based on photos of the meshes with a ruler on the netting and the image analysis facilities included in the FISHSELECT software tool [11, 12]. The entrance and exit diameter of the outer net part of the Excluder was kept open by two cylindrical kites made from PVC cloths (Fig 2). At the end of the 11 m square-meshed inner selection tube, a square PVC sail was mounted across the tube to partially block the water flow and force the fish to either actively bypass the sail or attempt swimming through the square meshes of the inner selection tube.

## Data collection

The total catch was estimated for each haul by the skipper inspecting the catch indicators in the Refrigerated-Salt-Water tanks (RSW). For each codend catch a fixed amount of 12 full baskets (approx. 340 kg in total), spread evenly across the pumping period, were sampled from the fish/water separator to make the sample taken representative for the entire catch. This fixed sample size of 12 baskets corresponded to an average total catch weight sampling fraction of 0.43% (Min = 0.14% and Max = 0.94%) across all 21 codend catches included in the analysis. For each haul separately, the ratio between the weight in the 12 baskets and the estimated total

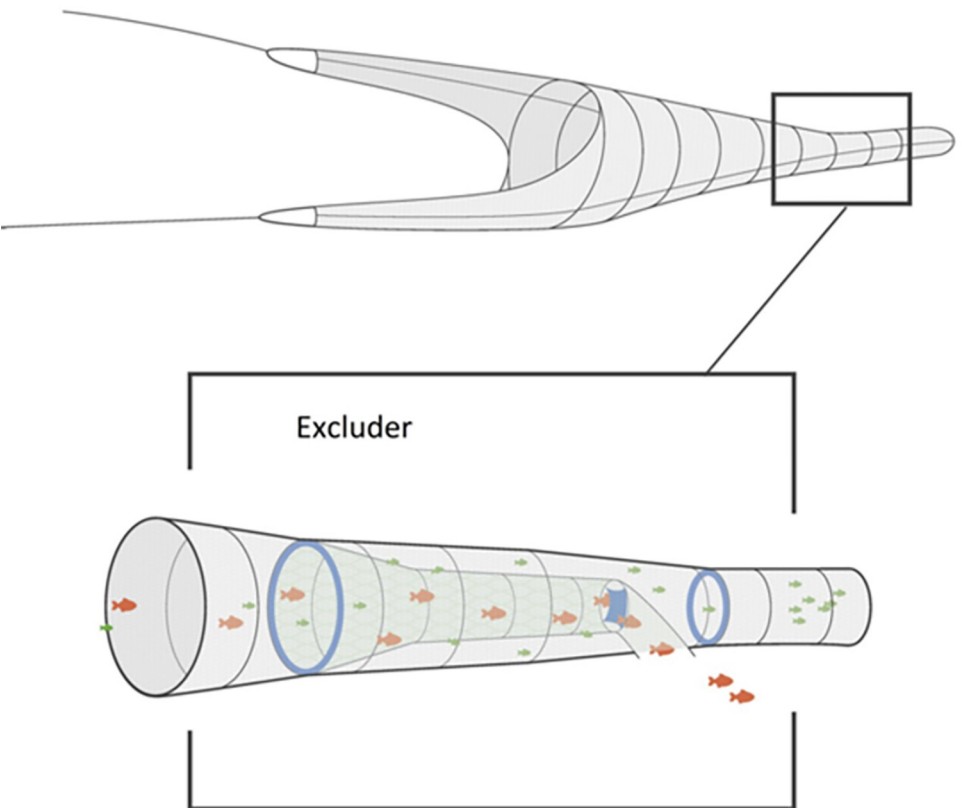

**Fig 2. Schematic representation of the Excluder's working principle.** The target species (lesser sandeel, symbolized by small, green-coloured fish) and the bycatch species (for instance mackerel, symbolized by larger, red-coloured fish) enter the trawl and the Excluder together. Inside the Excluder, the lesser sandeel swim through the 70mm square meshes of the inner selection tube and are guided to the codend. Mackerel cannot pass though the square meshes of the tube and are guided out of the Excluder.

catch was used as subsampling factor for the subsequent species composition analysis and species specific relative catch efficiency analysis. The full sample in the 12 baskets was sorted according to species, the number of individuals was counted for each species, and the total sample weight of each species was recorded for the species composition analysis. Further, for species being subjected to a length dependent relative catch efficiency analysis, the total length of each individual in the sample was measured and subsequently converted into count numbers for each one-centimetre-wide length classes. However, for lesser sandeel an additional sample level was introduced for the length measurements by conducting the measuring on only approx. 3.5 kg of the catch in the sample of this species due to the high number of individuals in the first sampling level. The resulting subsampling factors were used to account for additional uncertainty in species composition and species specific relative catch efficiency resulting from the subsampling procedure.

## Catch compositions analysis

Catch species compositions are often quantified by relative species dominance. Here, we use species dominance based both on count number of individuals and on weight to quantify the dominance of the individual species in the catch. The analysis was carried out for the two trawl configurations, without and with excluder, separately. We estimated the species dominance

for each species $i$ in each haul $j$ using the following equations [13]:

$$dn_{ij} = \frac{q_{ij} \times n_{ij}}{\sum_{i=1}^{K} \{q_{ij} \times n_{ij}\}}$$

$$dw_{ij} = \frac{q_{ij} \times n_{ij} \times \rho_{ij}}{\sum_{i=1}^{K} \{q_{ij} \times n_{ij} \times \rho_{ij}\}} \tag{1}$$

where $j$ represents the haul and $i$ is the species index (species rank) that was predefined. $n_{ij}$ is the number of individuals of the species $i$ being counted in the subsample in haul $j$. Parameter $\rho_{ij}$ is the average weight for one individual of species $i$ in haul $j$, whereas $q_{ij}$ is the fraction of species i in the catch being counted in haul $j$. $K$ is the total number of species considered. The relative species dominances $dn_{ij}$ and $dw_{ij}$ are numbers in the interval [0.0;1.0] that quantify the fraction of the total catch taken by species $i$ in haul $j$ based on respectively number of individuals and weight. Further the species dominance was also estimated averaged over hauls for each of the two trawl configurations separately by using:

$$dn_i = \frac{\sum_j \{q_{ij} \times n_{ij}\}}{\sum_{i=1}^{K} \sum_j \{q_{ij} \times n_{ij}\}}$$

$$dw_i = \frac{\sum_j \{q_{ij} \times n_{ij} \times \rho_{ij}\}}{\sum_{i=1}^{K} \sum_j \{q_{ij} \times n_{ij} \times \rho_{ij}\}} \tag{2}$$

In (2) the summation $j$ is over hauls with the specific trawl configuration.

The Efron percentile 95% confidence intervals (CIs; [14]) were used to provide the uncertainty of the values of dominance patterns obtained following the nested bootstrapping procedure described in Herrmann et al. [13]. This procedure enables estimation of the uncertainty around the dominance values at species level induced by the limited sample sizes at single haul without having to make any prior assumptions regarding the distributions in the hauls [13]. One thousand bootstrap repetitions were applied in the analysis.

Furthermore, the difference $\Delta d$ in species dominance $d$ in the standard trawl (control) and excluder trawl was estimated by:

$$\Delta d = d_{excluder} - d_{control} \tag{3}$$

By applying the technique described in Herrmann et al. [13], the CIs for Eq (3) were obtained based on separate bootstrap populations for $d_{control}$ and $d_{excluder}$. The significance was detected by inspecting if the CIs contained the value 0.0. If the 0.0 value was within the CIs, no significant difference was detected.

The catch composition analysis was conducted using the statistical software SELNET [12].

## Estimating relative catch efficiency between trawl with and without excluder

To quantify the effect by mounting the excluder in the trawl on the catch efficiency we conducted a length-dependent catch comparison and catch ratio analyses [15, 16]. As for the catch composition analysis above, we used the statistical analysis software SELNET to conduct the analysis. Using the catch data, we wanted to determine whether there was a significant difference in the catch efficiency between the trawl without (control, c) and the trawl with the excluder mounted (treatment, t). We also wanted to determine if a difference between the two trawl configurations could be related to fish size. Specifically, to assess the relative length-dependent catch efficiency effect of changing from trawl without to a trawl with excluder, we

used the method described in Herrmann et al. [17] based on comparing the catch data for hauls with and without excluder mounted in the trawl. Data were treated as unpaired [17], since the shift between trawl configurations were not done after each haul and therefore, we could not treat them as paired hauls data. The analysis was conducted species by species following the procedure described hereafter. Specifically, this method models the length-dependent catch comparison rate ($CC_l$) summed over hauls:

$$CC_l = \frac{\sum_{j=1}^{mt}\left\{\frac{nt_{lj}}{qt_j}\right\}}{\sum_{i=1}^{mc}\left\{\frac{nc_{li}}{qc_i}\right\} + \sum_{j=1}^{mt}\left\{\frac{nt_{lj}}{qt_j}\right\}} \tag{4}$$

where $mc$ and $mt$ are the number of hauls carried out with respectively the control trawl configuration and the treatment with the excluder mounted. $nc_{li}$ and $nt_{lj}$ are the numbers of fish measured in each length class $l$ (1-centimetre groups) for the control and the treatment trawl, respectively in tow $i$ and $j$. $qc_i$ and $qt_j$ are the related subsampling factors which is the fraction of the caught fish being length measured. However, the subsampling factors are, following Sistiaga et al. [18, 19], adjusted for the difference in towing time by multiplying the factor with the towing time for the specific tow and dividing with the towing time for the longest tow. The functional form catch comparison rate $CC(l,\mathbf{v})$ (the experimental being expressed by Eq 4), was obtained using maximum likelihood estimation by minimizing the following expression:

$$-\sum_l\left\{\sum_{i=1}^{mc}\left\{\frac{nc_{li}}{qc_i}\times ln(1.0 - CC(l,\mathbf{v}))\right\} + \sum_{j=1}^{mt}\left\{\frac{nt_{lj}}{qt_j}\times ln(CC(l,\mathbf{v}))\right\}\right\} \tag{5}$$

where $\mathbf{v}$ represents the parameters describing the catch comparison curve defined by $CC(l,\mathbf{v})$. The outer summation in the equation is the summation over the length classes $l$. When the catch efficiency of the control and treatment configuration of the trawl are equal, the expected value for the summed catch comparison rate would be $mc/(mc+mt)$. Therefore, this baseline can be applied to judge whether there is a difference in catch efficiency between the two trawls.

The experimental $CC_l$ was modelled by the function $CC(l,\mathbf{v})$, on the following form:

$$CC(l,\mathbf{v}) = \frac{exp(f(l,v_0,\ldots,v_k))}{1 + exp(f(l,v_0,\ldots,v_k))} \tag{6}$$

where $f$ is a polynomial of order $k$ with coefficients $v_0$ to $v_k$. The values of the parameters $\mathbf{v}$ describing $CC(l,\mathbf{v})$ are estimated by minimizing expression (5), which are equivalent to maximizing the likelihood of the observed data. We considered $f$ of up to an order of 4 with parameters $v_0$, $v_1$, $v_2$, $v_3$ and $v_4$. Leaving out one or more of the parameters $v_0\ldots v_4$ led to 31 additional models that were also considered as potential models for the catch comparison $CC(l,\mathbf{v})$. Among these models, estimations of the catch comparison rate were made using multimodel inference to obtain a combined model [17, 20]).

CI 95% limits for the catch comparison curves were estimated using a double bootstrapping method [17]. This bootstrapping method accounts for the uncertainty in the estimation resulting from tow variation in catch efficiency and availability of fish as well as uncertainty about the size structure of the catch for the individual hauls. By multi-model inference in each bootstrap iteration, the method also accounts for the uncertainty due to uncertainty in model selection. We performed 1,000 bootstrap repetitions and calculated the Efron 95% [14] CIs.

A length-integrated average value for the catch ratio was also estimated directly from the experimental catch data by:

$$CR_{average} = \frac{\frac{1}{mt} \sum_l \sum_{j=1}^{mt} \left\{ \frac{nt_{lj}}{qt_j} \right\}}{\frac{1}{mc} \sum_l \sum_{i=1}^{mc} \left\{ \frac{nc_{li}}{qc_i} \right\}} \tag{7}$$

where the outer summation covers the length classes in the catch during the experimental fishing period.

## Results

### Sea trials

We conducted 21 valid hauls using the single trawl setup during the fishing trials in May 2022 (Table 1). Ten hauls were without the excluder and eleven with the Excluder. Tow duration varied between 280 and 900 minutes. Two additional hauls (8 and 11) with too short duration (less than 30 minutes) to be considered, were left out of the analysis.

In total 9 different species were registered in the catches (Table 1): Lesser sandeel (*A. marinus*), Herring (*Clupea harengus*), Whiting (*Merlangius merlangus*), Grey gurnard (*Eutrigla gurnardus*), Mackerel (*Scomber scombrus*), Haddock (*Melanogrammus aeglefinus*), Allis shad (*Alosa alosa*), Cod (*Gadus morhua*) and European sprat (*Sprattus sprattus*). Catch amounts in the individual hauls spanned from 36 to 250 tonnes.

### Catch dominance analysis

The catch composition in individual hauls measured in number of fish showed that in general the bycatch rates were less than approx. 1% (Table 2). An exception was haul 3, where only

**Table 1. Date, position, gear type, duration, catch size, and towing speed of the 21 experimental hauls.**

| Tow ID | Date | Gear | Duration (minutes) | Catch (kg) | Tow speed (kn) |
|---|---|---|---|---|---|
| 1 | 11 May 2022 | Excluder | 390 | 36000 | 3.4 |
| 2 | 12 May 2022 | Excluder | 645 | 90000 | 4.3 |
| 3 | 13 May 2022 | Excluder | 900 | 48000 | 3.8 |
| 4 | 14 May 2022 | Control | 445 | 140000 | 3.5 |
| 5 | 14 May 2022 | Control | 280 | 50000 | 4.0 |
| 6 | 15 May 2022 | Excluder | 390 | 120000 | 3.6 |
| 7 | 15 May 2022 | Excluder | 325 | 50000 | 3.7 |
| 9 | 16 May 2022 | Excluder | 685 | 100000 | 3.7 |
| 10 | 17 May 2022 | Control | 495 | 60000 | 3.6 |
| 12 | 18 May 2022 | Control | 450 | 180000 | 3.6 |
| 13 | 19 May 2022 | Control | 360 | 100000 | 3.6 |
| 14 | 19 May 2022 | Excluder | 460 | 70000 | 3.6 |
| 15 | 20 May 2022 | Excluder | 420 | 180000 | 3.7 |
| 16 | 20 May 2022 | Excluder | 450 | 250000 | 3.7 |
| 17 | 21 May 2022 | Control | 500 | 145000 | 3.7 |
| 18 | 21 May 2022 | Control | 410 | 100000 | 3.7 |
| 19 | 22 May 2022 | Control | 535 | 115000 | 3.6 |
| 20 | 22 May 2022 | Control | 358 | 65000 | 3.7 |
| 21 | 23 May 2022 | Excluder | 530 | 125000 | 3.6 |
| 22 | 23 May 2022 | Excluder | 320 | 52000 | 3.5 |
| 23 | 24 May 2022 | Control | 360 | 60000 | 3.5 |

**Table 2. Species by species dominance in % of total catch for individual hauls based on number of fish caught.** Values in () represents 95% CIs.

| Tow ID | Lesser sandeel | Herring | Whiting | Grey gurnard | Mackerel | Haddock | Allis shad | Cod | European sprat |
|---|---|---|---|---|---|---|---|---|---|
| 1 | 99.95(99.92–99.98) | 0.00(0.00–0.01) | 0 | 0.02(0.00–0.04) | 0.02(0.01–0.05) | 0 | 0 | 0 | 0 |
| 2 | 98.91(98.73–99.06) | 1.05(0.89–1.23) | 0 | 0.00(0.00–0.01) | 0.04(0.02–0.07) | 0 | 0 | 0 | 0 |
| 3 | 92.25(91.00–93.43) | 7.63(6.47–8.89) | 0.00(0.00–0.02) | 0.03(0.01–0.06) | 0.09(0.05–0.14) | 0 | 0 | 0 | 0 |
| 4 | 99.88(99.84–99.91) | 0.02(0.01–0.04) | 0.01(0.00–0.02) | 0.01(0.00–0.01) | 0.01(0.00–0.01) | 0.07(0.05–0.10) | 0.00(0.00–0.01) | 0 | 0 |
| 5 | 99.20(99.06–99.31) | 0.02(0.01–0.04) | 0.09(0.06–0.12) | 0 | 0.00(0.00–0.01) | 0.69(0.59–0.82) | 0 | 0 | 0 |
| 6 | 99.86(99.82–99.89) | 0 | 0.07(0.05–0.10) | 0.00(0.00–0.01) | 0.01(0.00–0.02) | 0.05(0.03–0.07) | 0 | 0 | 0.00(0.00–0.01) |
| 7 | 99.90(98.88–99.93) | 0.02(0.01–0.03) | 0.02(0.01–0.04) | 0.01(0.00–0.02) | 0.00(0.00–0.01) | 0.02(0.01–0.04) | 0.00(0.00–0.01) | 0 | 0.01(0.00–0.02) |
| 9 | 99.60(99.50–99.69) | 0.03(0.01–0.05) | 0.20(0.14–0.28) | 0 | 0.05(0.02–0.08) | 0.01(0.00–0.02) | 0.12(0.07–0.17) | 0 | 0 |
| 10 | 98.87(98.64–99.06) | 0.10(0.05–0.15) | 0.37(0.27–0.48) | 0.04(0.01–0.08) | 0.10(0.05–0.15) | 0.48(0.37–0.61) | 0.02(0.00–0.05) | 0.02(0.00–0.05) | 0 |
| 12 | 99.89(99.83–99.94) | 0 | 0.01(0.00–0.03) | 0 | 0.04(0.01–0.08) | 0.06(0.02–0.11) | 0 | 0 | 0 |
| 13 | 99.89(99.85–99.93) | 0 | 0.03(0.01–0.05) | 0.01(0.00–0.03) | 0.02(0.00–0.04) | 0.06(0.03–0.09) | 0 | 0 | 0 |
| 14 | 99.96(99.94–99.98) | 0.02(0.01–0.04) | 0.01(0.00–0.02) | 0 | 0.01(0.00–0.01) | 0 | 0 | 0 | 0 |
| 15 | 99.96(99.94–99.98) | 0.02(0.01–0.04) | 0.02(0.00–0.03) | 0 | 0 | 0 | 0 | 0 | 0 |
| 16 | 99.95(99.92–99.98) | 0.05(0.02–0.08) | 0 | 0 | 0 | 0 | 0 | 0 | 0 |
| 17 | 99.91(99.86–99.94) | 0.09(0.06–0.13) | 0 | 0.01(0.00–0.02) | 0 | 0 | 0 | 0 | 0 |
| 18 | 99.93(99.89–99.96) | 0.07(0.04–0.11) | 0 | 0 | 0 | 0 | 0 | 0 | 0 |
| 19 | 99.90(99.85–99.93) | 0.06(0.04–0.09) | 0 | 0 | 0.03(0.02–0.06) | 0.01(0.00–0.02) | 0 | 0 | 0 |
| 20 | 99.66(99.57–99.73) | 0.09(0.05–0.13) | 0.10(0.06–0.15) | 0.00(0.00–0.01) | 0.08(0.04–0.12) | 0.07(0.04–0.11) | 0 | 0 | 0 |
| 21 | 99.58(99.48–99.66) | 0.42(0.34–0.51) | 0 | 0.01(0.00–0.02) | 0 | 0 | 0 | 0 | 0 |
| 22 | 99.94(99.90–99.96) | 0.06(0.03–0.09) | 0 | 0.01(0.00–0.02) | 0 | 0 | 0 | 0 | 0 |
| 23 | 99.89(99.83–99.94) | 0.08(0.04–0.13) | 0.03(0.00–0.05) | 0.01(0.00–0.02) | 0 | 0 | 0 | 0 | 0 |

92.25% of fish caught were lesser sandeel (the next lowest dominance was 98.87%). In haul 3 the bycatch was dominated by herring with 7.63%. No other bycatch species exceeded 0.7% in dominance in any haul when based on number of fish captured.

When measured in catch weight (Table 3) the dominance of the targeted lesser sandeel is slightly lower than when measured in number of fish. However, except from haul 3 (68.45%) and 5 (81.53%) its dominance is still between 91% to nearly 100%. In haul 3 herring was the most dominating bycatch species with 31.10% of the catch in weight. In haul 5 the most dominant bycatch species was haddock with 17.01% of the total catch.

**Table 3. Species by species dominance in % of total catch for individual hauls based on weight.** Values in () represents 95% CIs.

| Tow ID | Lesser sandeel | Herring | Whiting | Grey gurnard | Mackerel | Haddock | Allis shad | Cod | European sprat |
|---|---|---|---|---|---|---|---|---|---|
| 1 | 99.77(99.62–99.89) | 0.02(0.00–0.07) | 0 | 0.11(0.02–0.21) | 0.10(0.03–0.19) | 0 | 0 | 0 | 0 |
| 2 | 94.52(93.68–95.30) | 5.27(4.53–6.12) | 0 | 0.02(0.00–0.08) | 0.19(0.08–0.30) | 0 | 0 | 0 | 0 |
| 3 | 68.45(64.50–72.01) | 31.10(27.39–34.77) | 0.02(0.00–0.07) | 0.09(0.02–0.18) | 0.33(0.17–0.51) | 0 | 0 | 0 | 0 |
| 4 | 97.71(97.02–98.31) | 0.17(0.08–0.26) | 0.23(0.04–0.47) | 0.05(0.00–0.13) | 0.09(0.00–0.22) | 1.71(1.19–2.29) | 0.04(0.00–0.11) | 0 | 0 |
| 5 | 81.53(79.11–83.84) | 0.16(0.07–0.27) | 1.28(0.81–1,78) | 0 | 0.02(0.00–0.07) | 17.01(14.78–19.37) | 0 | 0 | 0 |
| 6 | 98.26(97.75–98.68) | 0 | 1.00(0.67–1.35) | 0.02(0.00–0.06) | 0.07(0.01–0.14) | 0.65(0.40–0.95) | 0 | 0 | 0.01(0.00–0.02) |
| 7 | 99.19(98.95–99.42) | 0.12(0.05–0.20) | 0.29(0.14–0.44) | 0.09(0.03–0.16) | 0.02(0.00–0.06) | 0.24(0.12–0.37) | 0.03(0.00–0.10) | 0 | 0.02(0.01–0.04) |
| 9 | 97.98(97.45–98.47) | 0.07(0.01–0.15) | 0.89(0.61–1.22) | 0 | 0.24(0.08–0.41) | 0.03(0.00–0.09) | 0.79(0.49–1.14) | 0 | 0 |
| 10 | 91.71(90.07–93.29) | 0.27(0.14–0.43) | 1.79(1.29–2.34) | 0.31(0.10–0.58) | 0.68(0.36–1.10) | 5.04(3.80–6.42) | 0.09(0.00–0.21) | 0.10(0.00–0.24) | 0 |
| 12 | 98.67(97.93–99.28) | 0 | 0.04(0.00–0.10) | 0 | 0.62(0.20–1.15) | 0.66(0.26–1.14) | 0 | 0 | 0 |
| 13 | 98.88(98.37–99.37) | 0 | 0.13(0.03–0.26) | 0.06(0.00–0.14) | 0.15(0.00–0.36) | 0.79(0.35–1.30) | 0 | 0 | 0 |
| 14 | 99.85(99.74–99.93) | 0.06(0.02–0.11) | 0.06(0.00–0.13) | 0 | 0.03(0.00–0.08) | 0 | 0 | 0 | 0 |
| 15 | 99.80(99.65–99.92) | 0.05(0.01–0.09) | 0.15(0.03–0.30) | 0 | 0 | 0 | 0 | 0 | 0 |
| 16 | 99.82(99.73–99.90) | 0.18(0.10–0.27) | 0 | 0 | 0 | 0 | 0 | 0 | 0 |
| 17 | 99.64(99.47–99.77) | 0.28(0.18–0.41) | 0 | 0.07(0.00–0.18) | 0 | 0 | 0 | 0 | 0 |
| 18 | 99.73(99.60–99.83) | 0.27(0.17–0.40) | 0 | 0 | 0 | 0 | 0 | 0 | 0 |
| 19 | 98.19(97.26–99.00) | 0.26(0.16–0.39) | 0 | 0 | 1.42(0.65–2.34) | 0.12(0.00–0.31) | 0 | 0 | 0 |
| 20 | 95.34(93.97–96.65) | 0.24(0.14–0.35) | 0.84(0.53–1.21) | 0.03(0.00–0.08) | 2.59(1.45–3.86) | 0.97(0.51–1.45) | 0 | 0 | 0 |
| 21 | 98.54(98.15–98.87) | 1.29(1.03–1.61) | 0 | 0.17(0.00–0.43) | 0 | 0 | 0 | 0 | 0 |
| 22 | 99.77(99.65–99.87) | 0.17(0.10–0.26) | 0 | 0.06(0.00–0.14) | 0 | 0 | 0 | 0 | 0 |
| 23 | 99.74(99.56–99.87) | 0.11(0.05–0.17) | 0.14(0.03–0.29) | 0.01(0.00–0.04) | 0 | 0 | 0 | 0 | 0 |

When the species dominance is averaged across hauls for the two trawl configurations (Table 4), lesser sandeel dominates the catch with more than 99% in fish number for both control and Excluder hauls, and with more than 96% of the catch weight for both configurations. The delta analysis revealed no significant difference in species contribution, except for haddock where the use of excluder reduces the dominance significantly from already low levels of 0.15% (numbers) and 2.68% (weight). Overall, the most dominant bycatch species was herring, when measured in weight, with approximatively 3% dominance, however with a large uncertainty (CI: 0.15–8.55).

**Table 4. Average species dominance in % for the 10 control and the 11 excluder hauls in number of fish and catch weight.** Values in () represents 95 CIs. Significant differences between configurations are marked in bold.

| Species | Control in number individuals | Excluder in number individuals | Delta Excluder-Control in number individuals | Control in weight | Excluder in Weight | Delta Excluder-Control in weight |
|---|---|---|---|---|---|---|
| Lesser sandeel | 99.73 (99.51–99.89) | 99.41 (98.45–99.91) | -0.32(-1.32–0.25) | 96.09 (91.52–98.81) | 96.39 (90.97–99.36) | 0.30(-6,03–5.73) |
| Herring | 0.05 (0.03–0.07) | 0.52 (0.04–1.49) | 0.47(-0.01–1.44) | 0.17 (0.10–0.25) | 3.09 (0.15–8.55) | 2.92(-0.05–8.36) |
| Whiting | 0.05 (0.01–0.10) | 0.03 (0.01–0.06) | -0.02(-0.08–0.03) | 0.44 (0.10–0.90) | 0.22 (0.04–0.47) | -0.22(-0.71–0.18) |
| Grey gurnard | 0.01 (0.00–0.01) | 0.01 (0.00–0.01) | 0.00(-0.01–0.01) | 0.05 (0.01–0.12) | 0.05 (0.02–0.09) | -0.01(-0.09–0.06) |
| Mackerel | 0.02 (0.01–0.04) | 0.01 (0.00–0.03) | -0.01(-0.03–0.02) | 0.54 (0.12–1.13) | 0.09 (0.03–0.16) | -0.46(-1.04- -0.04) |
| Haddock | 0.15 (0.03–0.33) | 0.01 (0.00–0.02) | **-0.14(-0.32- -0.01)** | 2.68 (0.47–6.74) | 0.08 (0.00–0.22) | **-2.59(-6.46- -0.32)** |
| Allis shad | 0.00 (0.00–0.00) | 0.01 (0.00–0.02) | 0.001(0.00–0.02) | 0.01 (0.00–0.04) | 0.08 (0.00–0.24) | 0.06(-0.03–0.23) |
| Cod | 0.00 (0.00–0.00) | 0.00 (0.00–0.00) | 0.00(0.00–0.00) | 0.01 (0.00–0.04) | 0.00 (0.00–0.00) | -0.01((-0.04–0.00) |
| European sprat | 0.00(0.00–0.00) | 0.00 (0.00–0.01) | 0.00(0.00–0.01) | 0.00 (0.00–0.00) | 0.00 (0.00–0.01) | 0.00(0.00–0.01) |

## Relative average catch efficiency

Summed over hauls and length classes the relative catch efficiency for lesser sandeel of the excluder trawl was estimated to be ≈98% of that of the control trawl (Table 5). However, confidence bands are wide (45–189%) implying that neither a considerable loss nor increase can be ruled out based on the collected experimental data. The same is implied for the bycatch species herring, whiting, grey gurnard, and mackerel whereas a significant reduction is observed for haddock (Table 5).

## Size dependent relative catch efficiency

The size dependent relative catch efficiency (Table 6 and Fig 3) provides a more detailed picture than the estimates summed over sizes. The p-value is below 0.05 for all species which could indicate poor model fit (Table 6). However, since the modelled catch comparison curve seem to follow the main trend in the experimental data well (Fig 3) we assume that the low p-values are a result of overdispersion in the experimental data. Therefore, we are confident in applying the modelled results for representing the relative catch efficiency between excluder and control trawl. For lesser sandeel, catch efficiency is not found to differ significantly except for sizes in the 20 cm length class, where there is a reduction with the excluder trawl. For the

**Table 5. Average catch ratio (Average CR) between the Excluder and control hauls in percentage.** Values in () represents 95% confidence bands. Values marked in bold are where catch efficiency differ significantly between excluder and control trawl.

| Species | CR Average (%) |
|---|---|
| Lesser sandeel | 97.75 (45.60–188.77) |
| Herring | 474.04 (92.96–1346.88) |
| Whiting | 100.43 (13.58–363.33) |
| Grey gurnard | 95.26 (12.27–305.81) |
| Mackerel | 54.77 (14.94–200.63) |
| Haddock | **9.27 (0.00–56.58)** |

**Table 6. Length dependent relative catch efficiency (excluder/control trawl) in %.** Values in () represents 95% confidence bands. Values marked in bold are where significant differences are obtained between excluder and control trawl regarding catch efficiency. The analysis was done by 1-centimetre groups, but due to space considerations the table only shows results for every second group.

| Length (cm) | Lesser sandeel | Herring | Whiting | Grey gurnard | Mackerel | Haddock |
|---|---|---|---|---|---|---|
| 6 | 0.06(-7.55−671.04) | ** | ** | ** | ** | ** |
| 8 | 2.84(-9.08−331.78) | ** | ** | ** | ** | ** |
| 10 | 31.15(-0.54−389.42) | 1333.63(18.12−568090.91) | ** | ** | ** | ** |
| 12 | 100.81(25.03−476.98) | 501.79(17.62−14071.59) | ** | ** | ** | ** |
| 14 | 129.80(59.14−294.16) | 198.21(8.25−1598.29) | ** | ** | ** | ** |
| 16 | 90.63(47.05−171.20) | 118.67(10.08−448.41) | 802.30(0.70−19157.26) | 75.13(2.35−1730.22) | 865.22(0.00−*) | 3.10(0.00−*) |
| 18 | 47.27(15.50−115.00) | 158.47(30.87−343.35) | 196.60(2.64−1884.73) | 264.28(74.43−12955.75) | 385.52(1.21−*) | 23.95(0.00−*) |
| 20 | **25.63(3.19−99.07)** | 708.11(34.58−1273.52) | 117.58(8.28−625.62) | 351.45(103.76−8812.15) | 209.23(67.43−*) | 53.59(0.00−3293.14) |
| 22 | 20.32(1.83−108.61) | 16192.43(28.27−55341.46) | 112.22(12.18−472.40) | 115.55(9.76−2860.68) | 128.96(65.40−*) | 41.48(0.00−786.41) |
| 24 | 33.51(7.60−294.43) | 2953891.07(25.14−*) | 109.65(14.02−430.30) | 6.02(0.00−176.78) | 82.46(3.17−*) | 13.53(0.00−133.42) |
| 26 | 164.99(37.38−*) | ** | 69.015(8.70−388.29) | **0.03(0.00−13.71)** | 49.03(0.00−*) | **2.31(0.00−18.26)** |
| 28 | 3522.36(48.41−*) | ** | 17.27(1.46−427.90) | **0.00(0.00−0.88)** | 23.83(0.00−*) | **0.26(0.00−3.05)** |
| 30 | ** | ** | 1.04(0.04−668.41) | **0.00(0.00−0.05)** | 8.16(0.00−793.90) | **0.03(0.00−0.60)** |
| 32 | ** | ** | 0.01(0.00−4589.34) | **0.00(0.00−0.01)** | **1.67(0.00−26.32)** | **0.00(0.00−0.19)** |
| 34 | ** | ** | 0.00(0.00−6385.94) | **0.00(0.00−0.00)** | **0.17(0.00−5.75)** | **0.00(0.00−0.13)** |
| 36 | ** | ** | ** | ** | ** | ** |
| 38 | ** | ** | ** | ** | ** | ** |
| p-value | <0.0001 | <0.0001 | <0.0001 | <0.0001 | <0.0001 | <0.0001 |
| DOF | 31 | 25 | 12 | 9 | 28 | 18 |
| Deviance | 414.68 | 124.97 | 110.92 | 105.17 | 349.37 | 437.05 |

bycatch species grey gurnard (> 24 cm), mackerel (> 30 cm), and haddock (> 24 cm) the capture probability is found to be significantly reduced for larger sizes of the species.

## Discussion

### Bycatch composition

In several trawl fisheries that target small-sized fish species, the unwanted bycatch of juveniles of other fish species, is a persistent issue [21–23]. In the sandeel fishing trips that we sampled here, the dominance of the target species was very high across both control and Excluder hauls in both numbers (approx. 99%) and weight (approx. 96%). Only one of the bycatch species exceeded 0.5% dominance in numbers (herring 0.52%, Excluder) and in weight only herring (3.1%, Excluder) and haddock (2.7%, control) exceeded 0.5%. Based on these bycatch percentages, the small-meshed trawl fishery for lesser sandeel can be considered a clean fishery without any obvious bycatch issues. The main reason for the small bycatch percentages and high dominance of target species in the catches is the behaviour of the sandeel, i.e., that they aggregate in large, dense schools with very little mixing of other species [24]. Even so, the very large catch volumes in the fishery (up to 250 tons in a single haul) means that even small bycatch percentages represent relatively large absolute values at the haul and trip level, and even more so at the fishery level, and this is also one of the key incentives for the industry-driven development of the Excluder.

### Selectivity and catch efficiency of the two systems

Summed over hauls and length classes the relative catch efficiency for the number of lesser sandeel of the excluder trawl was estimated to be approx. 98% of that of the control trawl, but with

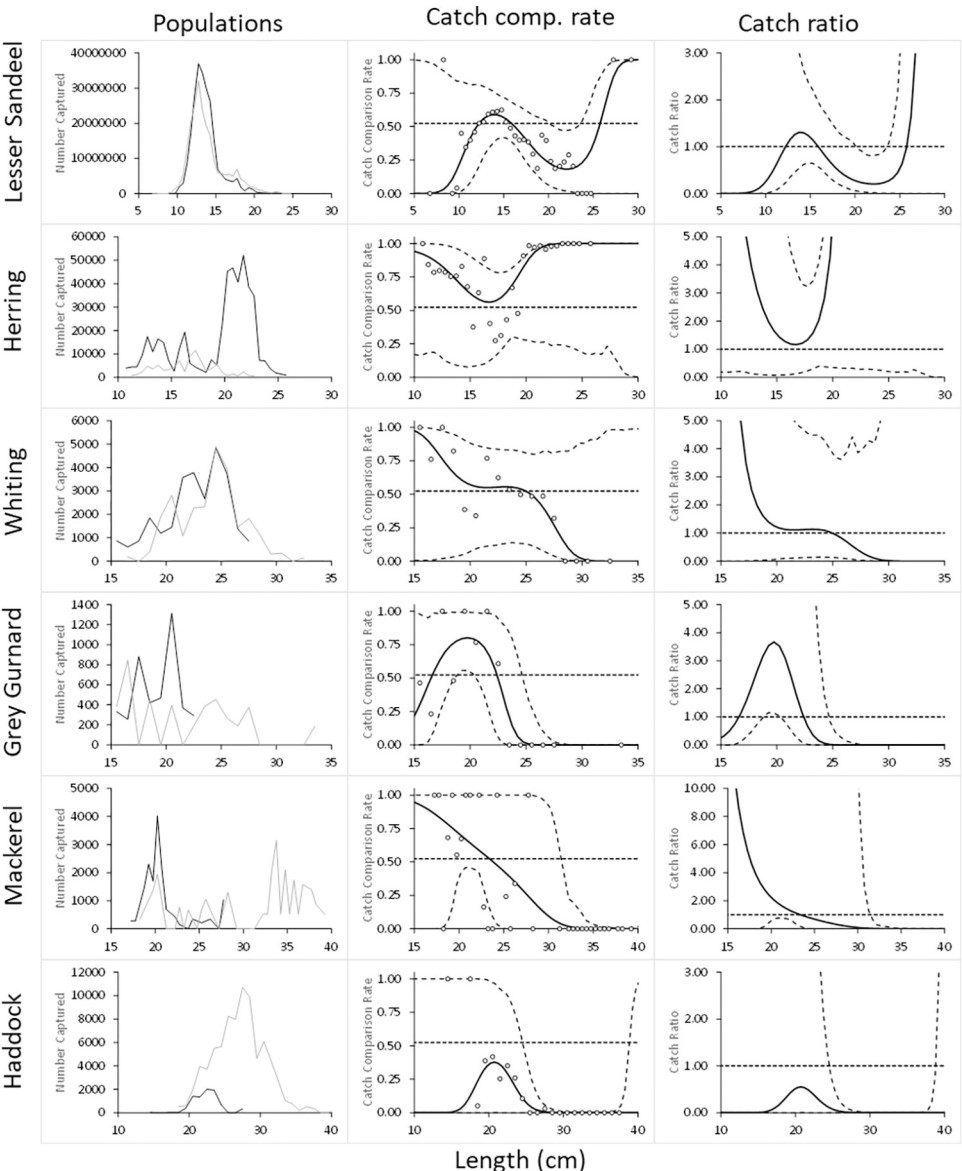

**Fig 3. Relative catch efficiency for main species of Excluder and control trawl.** The left-hand figure-panel shows the length frequency distributions of fish captured by the trawl with the Excluder (black curve) and the trawl without (grey curve). The middle figure-panel shows the modelled catch comparison rate of the two gear configurations (black line) with 95% CIs (black stippled curves). Circles represent the experimentally observed rate. The right-hand figure-panel shows the estimated catch ratio curve of the two gear configurations (black curve) with 95% CIs (black stippled curves). The grey stippled lines at 0.52 (= 11/ (11+10)) in the middle panels and 1.0 in the right panels represent the point at which both gears have an equal catch efficiency.

wide confidence bands (approx. 45–189%), implying that neither a considerable loss nor increase can be ruled out based on the collected experimental data. In a similar study comparing the use of an Excluder with a traditional rigid sorting grid in the Norway pout fishery, a 32% increase in target species catch efficiency was estimated [10]. Wide confidence bands also applied for the relative catch efficiency estimates of the bycatch species herring, whiting, grey gurnard, and mackerel, whereas overall haddock catch efficiency is significantly lower in the Excluder compared to the control.

The analyses of size-dependent relative catch efficiency of the two gear types confirmed almost identical sandeel catch efficiencies of the two gears, except for a significantly lower efficiency for sandeel in the 20 cm length class in the Excluder trawl. For the bycatch species grey gurnard (>24 cm), mackerel (>30 cm), and haddock (>24 cm) the capture probability is found to be significantly lower for larger sizes of the species in the Excluder. For other bycatch species the results were inconclusive due to wide confidence bands, affected by low bycatch numbers during the trials.

It was not unexpected that by adding a new sorting device (Excluder) on top of the existing selective system (trawl and codend), the catch efficiency for the target species as well as the bycatch species was reduced, and any other outcome of the experimental fishery would have been controversial. In the trials, significant reductions were limited to one centimetre group of the target species (the 20 cm length class), whereas the overall capture efficiency for sandeel was not significantly affected by adding the excluder. This outcome, together with a significant reduction in the overall catch efficiency for haddock and for the larger sizes of grey gurnard and mackerel, clearly highlights the potential of the Excluder as a bycatch reduction tool in the sandeel trawl fishery.

The very limited loss of target species also supports that one of the key objectives of the Excluder was achieved; that also very large volumes (pulses) of sandeel and bycatch species should be able to pass through the sorting device without blocking it. The key design principle behind this achievement is likely the large contact area available for the selection process in the Excluder. In a previous study in a comparable fishery [10] the sorting area of the experimental Excluder was roughly 15 times larger than the sorting area of the experimental grid, which is likely to significantly reduce the risk of clogging and related loss of function.

## Methodological improvements

Ideally the Excluder and standard trawl would have been deployed alternately one haul at a time, which would have minimized the temporal and spatial variation in the fish populations being trawled and have allowed for a paired hauls data approach to the analyses. However, due to the size of the gear, the installation and removal of the Excluder was a very time-consuming task, which was only possible to perform on a less regular basis. Hence the grouped distribution of Excluder and standard trawl hauls (typically 2 or 3 of each between shifts), and the less statistically robust unpaired data approach. Another characteristic of the sandeel fishery that complicated the experimental data collection, was that the very large gear and catch size did not allow weighing of all the catch. Instead, each codend catch was estimated to the nearest 1000 kg by the skipper based on the catch volume markers on the sides of the RSW-tanks. This procedure is more crude and more subject to uncertainty than exact measurements, but the approach is similar to how total catch weights are estimated and reported (legally binding) in the official logbooks, which adds confidence to the methodology and decreases the likelihood of any unintended bias.

## Excluder handling and safety

The Excluder has the advantage of being made of flexible material, enabling it to be reeled directly on the net drum together with the trawl. This ease of handling is particularly advantageous in high-volume fisheries, where the operation safety and handling time challenges of rigid sorting grids experienced in other trawl fisheries [25] are likely more pronounced. Furthermore, if implemented, the ease of handling of the Excluder would likely also increase compliance in the fishery, as compatibility between regulations and traditional fishing practices (as well as the efficacy of imposed regulations) has been shown to facilitate acceptance [26].

### Implications for the sandeel fishery

The main objective of the study was to develop and test an easy-handled, high-safety, netting-based sorting system for the sandeel fishery, which is capable of minimizing unintended bycatch while efficiently retaining the target species. With the Excluder system and the results of the sea trials this aim has been achieved, and the technological basis of a more selective sandeel fishery is in place. Furthermore, this project aim was achieved in a multidisciplinary approach involving academic expertise, gear manufacturers and the commercial fishery, which enhances the sense of industry buy-in and the likelihood of uptake in the fishing fleet. Assuming that the species and size distributions encountered during the experimental fishing are representative of those in the wider commercial fishery, a widespread implementation of the Excluder would lead to a reduction in bycatches of larger mackerel, grey gurnard, and haddock, and significantly improve the sustainability of the fishery.

### Perspectives

The Excluder used in this sandeel experiment is a first design and therefore further improvement to increase its efficiency, both in terms of bycatch reduction and target species efficiency, could be possible. An elongation of the inner selection tube to increase the sorting area of the Excluder is an obvious first step to explore. The selective properties of the Excluder could likely also be improved by optimizing the mesh size of the inner section, to optimize the trade-off between reduction of unwanted bycatch and loss of target species. By either increasing or decreasing the mesh opening the current balance can be shifted to either side, potentially providing a better trade-off result. The tested Excluder was designed specifically for the small-meshed sandeel fishery in the North Sea area with the objective of retaining this small-sized target species and sorting out larger ones, but it also represents a potential solution to similar sorting issues, as well as to grid-handling and clogging problems in other trawl fisheries. In the North east Atlantic other obvious sorting issues to explore are the unwanted saithe (Pollachius virens) bycatches in fisheries targeting herring and the bycatches of larger pelagic species, such as salmon, in the fisheries targeting sprat (sprattus sprattus) in the Baltic Sea. Globally there is likely also a substantial number of trawl fisheries with similar sorting issues that would benefit from introducing tailored netting-based Excluder systems, in some cases to replace existing grid-based systems.

### Supporting information

**S1 File. Data for individual hauls.** The raw data underlying the analysis consist of count data for number of individuals of all species caught with respectively the Excluder codend and the Grid codend, for each size class (length). The count numbers are provided for each sample of each haul.
(ZIP)

### Acknowledgments

We wish to thank the crews and teams of "S205 Ceton", the Flume tank in Hirtshals, and Jonathan Stounberg for their cooperation and assistance in the project, and not least Greenline Fishing Gear (http://www.greenlinefishinggear.com) and Tor-Mo Trawl ApS (http://www.tormotrawl.dk/) for developing and producing the tested Excluder.

### Author Contributions

**Conceptualization:** Ole R. Eigaard, Claus R. Sparrevohn, Mathias Søgaard, Bent Herrmann.

**Data curation:** Ole R. Eigaard, Claus R. Sparrevohn, Mathias Søgaard, Bent Herrmann.

**Formal analysis:** Ole R. Eigaard, Claus R. Sparrevohn, Mathias Søgaard, Bent Herrmann.

**Funding acquisition:** Ole R. Eigaard, Claus R. Sparrevohn.

**Investigation:** Ole R. Eigaard, Claus R. Sparrevohn, Mathias Søgaard, Bent Herrmann.

**Methodology:** Ole R. Eigaard, Claus R. Sparrevohn, Bent Herrmann.

**Project administration:** Ole R. Eigaard, Claus R. Sparrevohn.

**Resources:** Ole R. Eigaard, Claus R. Sparrevohn.

**Software:** Bent Herrmann.

**Supervision:** Ole R. Eigaard.

**Validation:** Ole R. Eigaard, Claus R. Sparrevohn, Mathias Søgaard, Bent Herrmann.

**Visualization:** Ole R. Eigaard, Bent Herrmann.

**Writing – original draft:** Ole R. Eigaard, Bent Herrmann.

**Writing – review & editing:** Ole R. Eigaard, Claus R. Sparrevohn, Mathias Søgaard, Bent Herrmann.

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
