## [Decision Letter · Decision Letter 0]

11 Jun 2024

PONE-D-24-11630Effect on catch efficiency and bycatch by introducing an Excluder device in the trawl fishery for lesser sandeel *(Ammodytes marinus)**PLOS ONE*

Dear Dr. Eigaard,

Thank you for submitting your manuscript to PLOS ONE. After careful consideration, we feel that it has merit but does not fully meet PLOS ONE’s publication criteria as it currently stands. Therefore, we invite you to submit a revised version of the manuscript that addresses the points raised during the review process.

We look forward to receiving your revised manuscript.

Kind regards,

Abdul Azeez Pokkathappada, Ph.D.

Academic Editor

PLOS ONE

Journal Requirements:

2. To comply with PLOS ONE submissions requirements, in your Methods section, please provide additional information regarding the experiments involving animals and ensure you have included details on (a) methods of sacrifice, (b) methods of anesthesia and/or analgesia, and (c) efforts to alleviate suffering.

This study was financed by the Ministry of Food, Agriculture and Fisheries of Denmark through the GUDP (Green Development and Demonstration Program) in the project ‘MiniMakS’ (34009-20-1674). We also wish to thank the crews and teams of “S205 Ceton”, the Flume tank in Hirtshals, and Jonathan Stounberg for their cooperation and assistance in the project, and not least Greenline Fishing Gear (http://www.greenlinefishinggear.com) and Tor-Mo Trawl ApS (http://www.tormotrawl.dk/) for developing and producing the tested Excluder. 

We wish to thank the Ministry of Food, Agriculture, and Fisheries of Denmark (https://fvm.dk/) for funding the work presented here through the GUDP (Green Development and Demonstration Program) in the project ‘MiniMakS’ (34009-20-1674). The funder did not play any role in the study design, data collection and analysis, decision to publish, or preparation of the manuscript.

5. We note that Figure 1 in your submission contain map images which may be copyrighted. All PLOS content is published under the Creative Commons Attribution License (CC BY 4.0), which means that the manuscript, images, and Supporting Information files will be freely available online, and any third party is permitted to access, download, copy, distribute, and use these materials in any way, even commercially, with proper attribution. For these reasons, we cannot publish previously copyrighted maps or satellite images created using proprietary data, such as Google software (Google Maps, Street View, and Earth). For more information, see our copyright guidelines: http://journals.plos.org/plosone/s/licenses-and-copyright.

We require you to either present written permission from the copyright holder to publish these figures specifically under the CC BY 4.0 license, or remove the figures from your submission:

Reviewers' comments:

Reviewer's Responses to Questions

**Comments to the Author**

1. Is the manuscript technically sound, and do the data support the conclusions?

Reviewer #1: Yes

Reviewer #2: Yes

2. Has the statistical analysis been performed appropriately and rigorously? 

Reviewer #1: Yes

Reviewer #2: Yes

3. Have the authors made all data underlying the findings in their manuscript fully available?

Reviewer #1: Yes

Reviewer #2: Yes

4. Is the manuscript presented in an intelligible fashion and written in standard English?

Reviewer #1: Yes

Reviewer #2: Yes

5. Review Comments to the Author

**Reviewer #1: **The use of soft BRD Excluder significantly reduced the bycatch ratio above certain sizes while did not affected the catch efficiency for lesser sandeel. Although it is not mandatory to be used in the local sandeel trawl fishery, its soft-netting, catching effectiveness and easy-operation properties indicated its potential to be used in other commercial fisheries and to be an alternative for rigid BRD. The results was significant for research and fishing practice, and the paper was well written and data analysis was reasonable. It is recommended to be accepted with minor revisions below:

Table 1. hauling positions (latitude and longitude data) could be removed since they are already presented in Figure 1.

Too many tables in the paper. Could Table 2 and 3 be transferred to figures? It is easy to see the difference between control and Exclude in a figure.

I want to see a discussion section about the mess block of the Excluder. A larger volume of sandeel and some bycatch species need to bypass the device (in a short time?). Are the mesh blocked by the passed fishes and therefore affect the effectiveness of the device although the towing speed was not so high?

**Reviewer #2:** The manuscript evaluates the effect of an Excluder device on catch efficiency and bycatch reduction in the trawl fishery for lesser sandeel. The study involves experimental sea trials comparing standard trawl configuration with and without the Excluder. Results indicate that the Excluder significantly reduces bycatch of non-target species such as haddock, mackerel, and grey gurnard without substantially affecting the catch efficiency of the target species.

The study addresses a relevant issue in fisheries management by investigating the potential of the Excluder device to reduce bycatch in a targeted sandeel fishery. The manuscript is well-structured, and the methods and analyses are generally sound. However, there are a few minor corrections needed to improve clarity and accuracy.

Minor Corrections:

Line 72:The scientific name “Ammodytes marinus” should be abbreviated to “A. marinus” in its second occurrence within the parentheses.

Line 80:Please add the scientific name for Norway pout, which is “Trisopterus esmarkii.”

Experimental objectives This innovative design is particularly beneficial for net drum users, as it ensures the Excluder can be easily reeled onto net drums, enhancing both operational efficiency and safety.

Data collection: In summary, the data collection method described has several critical flaws that need to be addressed to enhance its scientific validity. Reliance on subjective skipper estimates, lack of clarity on sample collection timing, and inconsistent sampling fractions are significant issues. Clear guidelines for sub-sampling and a thorough explanation of measurement and raising techniques are necessary. Addressing these concerns will improve the accuracy and reliability of the data collected, ensuring more robust scientific outcomes.

Line 159: In Equation 1, the variables d_n and d_w need to be defined in the text. Please provide definitions for these terms to ensure clarity.

Overall, the manuscript provides valuable insights into bycatch reduction technologies and their potential applications in sustainable fisheries management. With these minor corrections, the manuscript is suitable for publication in PLOS ONE.

Recommendation:

Minor Revisions: The manuscript should be revised to address the specific corrections outlined above. With these minor revisions, it is suitable for publication.

6. PLOS authors have the option to publish the peer review history of their article (what does this mean?). If published, this will include your full peer review and any attached files.

Reviewer #1: No

Reviewer #2: No

---

## [Author Response · Author response to Decision Letter 0]

19 Jun 2024

[PONE-D-24-11630] Revision required

Comments from Academic editor:

Journal Requirements:

Specific comments 

### The manuscript has been revised to meet PLOS ONE's style requirements

2. To comply with PLOS ONE submissions requirements, in your Methods section, please provide additional information regarding the experiments involving animals and ensure you have included details on (a) methods of sacrifice, (b) methods of anesthesia and/or analgesia, and (c) efforts to alleviate suffering. 

### The requested information has been added in the Methods section

This study was financed by the Ministry of Food, Agriculture and Fisheries of Denmark through the GUDP (Green Development and Demonstration Program) in the project ‘MiniMakS’ (34009-20-1674). We also wish to thank the crews and teams of “S205 Ceton”, the Flume tank in Hirtshals, and Jonathan Stounberg for their cooperation and assistance in the project, and not least Greenline Fishing Gear (http://www.greenlinefishinggear.com) and Tor-Mo Trawl ApS (http://www.tormotrawl.dk/) for developing and producing the tested Excluder. 

We wish to thank the Ministry of Food, Agriculture, and Fisheries of Denmark (https://fvm.dk/) for funding the work presented here through the GUDP (Green Development and Demonstration Program) in the project ‘MiniMakS’ (34009-20-1674). The funder did not play any role in the study design, data collection and analysis, decision to publish, or preparation of the manuscript.

### We have deleted the funding information from the manuscript. This also means that the current information and wording in our Funding Statement is correct.

### The requested information has been added in the Methods section.

5. We note that Figure 1 in your submission contains map images which may be copyrighted. All PLOS content is published under the Creative Commons Attribution License (CC BY 4.0), which means that the manuscript, images, and Supporting Information files will be freely available online, and any third party is permitted to access, download, copy, distribute, and use these materials in any way, even commercially, with proper attribution. For these reasons, we cannot publish previously copyrighted maps or satellite images created using proprietary data, such as Google software (Google Maps, Street View, and Earth). For more information, see our copyright guidelines: http://journals.plos.org/plosone/s/licenses-and-copyright. We require you to either present written permission from the copyright holder to publish these figures specifically under the CC BY 4.0 license, or remove the figures from your submission: a. You may seek permission from the original copyright holder of Figure 1 to publish the content specifically under the CC BY 4.0 license. 

“I request permission for the open-access journal PLOS ONE to publish XXX under the Creative Commons Attribution License (CCAL) CC BY 4.0 (http://creativecommons.org/licenses/by/4.0/). Please be aware that this license allows unrestricted use and distribution, even commercially, by third parties. Please reply and provide explicit written permission to publish XXX under a CC BY license and complete the attached form.” Please upload the completed Content Permission Form or other proof of granted permissions as an "Other" file with your submission. In the figure caption of the copyrighted figure, please include the following text: “Reprinted from [ref] under a CC BY license, with permission from [name of publisher], original copyright [original copyright year].” b. If you are unable to obtain permission from the original copyright holder to publish these figures under the CC BY 4.0 license or if the copyright holder’s requirements are incompatible with the CC BY 4.0 license, please either i) remove the figure or ii) supply a replacement figure that complies with the CC BY 4.0 license. Please check copyright information on all replacement figures and update the figure caption with source information. If applicable, please specify in the figure caption text when a figure is similar but not identical to the original image and is therefore for illustrative purposes only. The following resources for replacing copyrighted map figures may be helpful: USGS National Map Viewer (public domain): http://viewer.nationalmap.gov/viewer/ The Gateway to Astronaut Photography of Earth (public domain): http://eol.jsc.nasa.gov/sseop/clickmap/Maps at the CIA (public domain): https://www.cia.gov/library/publications/the-world-factbook/index.html and https://www.cia.gov/library/publications/cia-maps-publications/index.html

NASA Earth Observatory (public domain): http://earthobservatory.nasa.gov/Landsat: http://landsat.visibleearth.nasa.gov/

USGS EROS (Earth Resources Observatory and Science (EROS) Center) (public domain): http://eros.usgs.gov/# Natural Earth (public domain): http://www.naturalearthdata.com/

##### Figure 1 in the manuscript is not a [map/satellite] image, but has been produced by the authors in ArcGIS, based on UTM position data from the experimental vessel.

Consequently, there is no issue with copyright.

6. Please review your reference list to ensure that it is complete and correct. If you have cited papers that have been retracted, please include the rationale for doing so in the manuscript text or remove these references and replace them with relevant current references. Any changes to the reference list should be mentioned in the rebuttal letter that accompanies your revised manuscript. If you need to cite a retracted article, indicate the article’s retracted status in the References list and also include a citation and full reference for the retraction notice.

### The reference list has been reviewed. No changes were made.

Review Comments to the Author

Reviewer #1: The use of soft BRD Excluder significantly reduced the bycatch ratio above certain sizes while did not affect the catch efficiency for lesser sandeel. Although it is not mandatory to be used in the local sandeel trawl fishery, its soft-netting, catching effectiveness and easy-operation properties indicated its potential to be used in other commercial fisheries and to be an alternative for rigid BRD. The results were significant for research and fishing practice, and the paper was well written and data analysis was reasonable. It is recommended to be accepted with minor revisions below:

Specific comments

1. Table 1. hauling positions (latitude and longitude data) could be removed since they are already presented in Figure 1 

### Positions removed from Table 1

2. Too many tables in the paper. Could Table 2 and 3 be transferred to figures? It is easy to see the difference between control and Excluder in a figure. 

### We have carefully considered this suggestion from the reviewer. However, the results presented in Table 2 and Table 3 are not available from any of the figures shown in the manuscript. Therefore, as these results are important and central for the paper we prefer keeping the existing Table 2 and 3 in the manuscript, also considering that the total number of Tables is only 6.

3. I want to see a discussion section about the mesh block of the Excluder. A larger volume of sandeel and some bycatch species need to bypass the device (in a short time?). Are the mesh blocked by the passed fishes and therefore affect the effectiveness of the device although the towing speed was not so high? 

### A discussion section has been added as suggested.

Reviewer #2: The manuscript evaluates the effect of an Excluder device on catch efficiency and bycatch reduction in the trawl fishery for lesser sandeel. The study involves experimental sea trials comparing standard trawl configuration with and without the Excluder. Results indicate that the Excluder significantly reduces bycatch of non-target species such as haddock, mackerel, and grey gurnard without substantially affecting the catch efficiency of the target species. The study addresses a relevant issue in fisheries management by investigating the potential of the Excluder device to reduce bycatch in a targeted sandeel fishery. The manuscript is well-structured, and the methods and analyses are generally sound. However, there are a few minor corrections needed to improve clarity and accuracy.

Specific comments

1. Line 72: The scientific name “Ammodytes marinus” should be abbreviated to “A. marinus” in its second occurrence within the parentheses. 

### The manuscript has been revised according to the comment.

2. Line 80: Please add the scientific name for Norway pout, which is “Trisopterus esmarkii.” 

### The manuscript has been revised according to the comment.

3. Experimental objectives This innovative design is particularly beneficial for net drum users, as it ensures the Excluder can be easily reeled onto net drums, enhancing both operational efficiency and safety. 

### The manuscript has been revised according to the comment.

4. Data collection: In summary, the data collection method described has several critical flaws that need to be addressed to enhance its scientific validity. Reliance on subjective skipper estimates, lack of clarity on sample collection timing, and inconsistent sampling fractions are significant issues. Clear guidelines for sub-sampling and a thorough explanation of measurement and raising techniques are necessary. Addressing these concerns will improve the accuracy and reliability of the data collected, ensuring more robust scientific outcomes. 

### We agree with the reviewer that we did not describe in sufficient detail how the data collection was conducted and how we have accounted for the uncertainty due to the sampling procedure and the variation in sampling fractions. We have now described this in more detail in the revised manuscript.

5. Line 159: In Equation 1, the variables d_n and d_w need to be defined in the text. Please provide definitions for these terms to ensure clarity. 

### Thanks for the comment, the reviewer is correct. We should explain better. Therefore, have added text that explains in more detail what these performances measures express.

---

## [Editor Report · Decision Letter 1]

24 Jun 2024

Effect on catch efficiency and bycatch by introducing an Excluder device in the trawl fishery for lesser sandeel *(Ammodytes marinus)*

*PONE-D-24-11630R1*

*Dear Dr. Eigaard,*

*We’re pleased to inform you that your manuscript has been judged scientifically suitable for publication and will be formally accepted for publication once it meets all outstanding technical requirements.*

*Within one week, you’ll receive an e-mail detailing the required amendments. When these have been addressed, you’ll receive a formal acceptance letter and your manuscript will be scheduled for publication.*

*An invoice will be generated when your article is formally accepted. Please note, if your institution has a publishing partnership with PLOS and your article meets the relevant criteria, all or part of your publication costs will be covered. Please make sure your user information is up-to-date by logging into Editorial Manager at Editorial Manager® and clicking the ‘Update My Information' link at the top of the page. If you have any questions relating to publication charges, please contact our Author Billing department directly at authorbilling@plos.org.*

*If your institution or institutions have a press office, please notify them about your upcoming paper to help maximize its impact. If they’ll be preparing press materials, please inform our press team as soon as possible -- no later than 48 hours after receiving the formal acceptance. Your manuscript will remain under strict press embargo until 2 pm Eastern Time on the date of publication. For more information, please contact onepress@plos.org.*

*Kind regards,*

*Abdul Azeez Pokkathappada, Ph.D.*

Academic Editor

*PLOS ONE*

* *

*Additional Editor Comments (optional):*

*The authors are kindly requested to replace all the figures in the manuscript with at least 300 dpi quality, so that readers can view them clearly and without strain.*

* *
---

## [Editor Report · Acceptance letter]

26 Jun 2024

PONE-D-24-11630R1 

PLOS ONE

Dear Dr. Eigaard, 

I'm pleased to inform you that your manuscript has been deemed suitable for publication in PLOS ONE. Congratulations! Your manuscript is now being handed over to our production team.

Kind regards, 

on behalf of

Dr. Abdul Azeez Pokkathappada 

Academic Editor

PLOS ONE